# Performance Assessment of Self-Healing Polymer-Modified Bitumens by Evaluating the Suitability of Current Failure Definition, Failure Criterion, and Fatigue-Restoration Criteria

**DOI:** 10.3390/ma16062488

**Published:** 2023-03-21

**Authors:** Songtao Lv, Dongdong Ge, Ziyang Wang, Jinping Wang, Jing Liu, Zihao Ju, Xinghai Peng, Xiyan Fan, Shihao Cao, Dingyuan Liu, Wenhui Zhang, Milkos Borges Cabrera

**Affiliations:** School of Traffic and Transportation Engineering, Changsha University of Science & Technology, Changsha 410114, China; lst@csust.edu.cn (S.L.); dge1@csust.edu.cn (D.G.); 17307405079@163.com (Z.W.); Wangjp@stu.csust.edu.cn (J.W.); lj0030@stu.csust.edu.cn (J.L.); jzh@stu.csust.edu.cn (Z.J.); pengxinghsi@stu.csust.edu.cn (X.P.); fxy@stu.csust.edu.cn (X.F.); csh@stu.csust.edu.cn (S.C.); nf@stu.csust.edu.cn (D.L.); whzz@stu.csust.edu.cn (W.Z.)

**Keywords:** bitumen fatigue performance, bitumen fatigue failure, failure definition, failure criterion, self-restoration, self-healing

## Abstract

Fatigue cracking is a common form of flexible pavement distress, which generally starts and spreads through bitumen. To address this issue, self-healing elastomer (SHE) modified bitumens were elaborated to assess whether these novel materials can overcome the neat asphalt (NA) fatigue performance and whether the current failure definition, failure criterion, and fatigue-restoration criteria can fit their performance. All bitumens were subjected to short-term and long-term aging. Linear amplitude sweep (LAS) test, LAS with rest period (LASH), and simplified viscoelastic-continuum-damage (S-VECD) model were utilized to appraise the behavior of the mentioned bitumens. The results showed that maximum stored pseudo-strain energy (PSE) and tau (τ) × N (number of loading cycles) failure definitions exhibited high efficiency to accommodate the fatigue life of NA and SHE-modified bitumens. Both failure criteria identified that SHE-modified bitumen (containing 1% of SHE) showed the highest increment of fatigue performance (67.1%) concerning NA. The failure criterion based on total released PSE, in terms of the area under the released PSE curve, was the only failure concept with high efficiency (R^2^ up to 0.999) to predict asphalt binder fatigue life. As well, the current framework to evaluate bitumen self-restoration failed to fully accommodate asphalt binder behavior, because bitumen with higher restoration could not exhibit greater fatigue performance. Consequently, a new procedure to assess this property including fatigue behavior was proposed, showing consistent results, and confirming that SHE-modified bitumen (containing 1% of SHE) exhibited the highest increment of fatigue performance (154.02%) after application of the rest period. Hence, the optimum SHE content in NA was 1%. Furthermore, it was found that a greater number of loading cycles to failure ($N_{f}$) did not ensure better fatigue performance and stored PSE influenced the bitumen fatigue behavior.

## 1. Introduction

Fatigue cracking represents one of the principal distresses in bituminous pavements, and cyclic traffic loading causes it [1]. Pavement fatigue cracking prediction still represents a challenge for researchers and scholars. A full understanding of fatigue cracking evolution is a task that needs to be accomplished [2,3,4]. The fatigue cracking phenomenon in asphalt mixture generally starts and spreads through bitumen; hence the asphalt pavement fatigue performance significantly depends on the asphalt binder fatigue properties [5,6,7]. At present, numerous parameters can be utilized to assess (or have influence on) bitumen fatigue performance, and three of the most important are: failure definition, failure criterion, and self-healing. Linear Amplitude Sweep (LAS) test and simplified viscoelastic continuum damage (S-VECD) theory are utilized to obtain these parameters. AASHTO TP 101 explains the LAS test, and this procedure is considered an effective and time-saving process to assess the asphalt binder fatigue performance. The LAS test experimental results have to be processed by utilizing the S-VECD model, which allows for obtaining the damage characteristics curve (DCC). This graph constitutes the relationship between material integrity (C) (also known as pseudo-stiffness and normalized dynamic shear modulus) and an internal state variable representing damage intensity (S) derived considering Schapery’s work potential theory for damage evolution, and its formula is as follows [8,9]:(1)$$\frac{dS}{d\vartheta }={\left(-\frac{\partial W^{R}}{\partial S}\right)}^{\alpha }$$where ϑ and $W^{R}$ are reduced time and pseudo-strain energy, resepectively.

Failure definition identifies the failure occurrence in a specific material and consequently establishes its fatigue life ($N_{f}$). Failure definition ought to be carefully chosen, with an exhaustive and balanced analysis of experimental results, theoretical framework, and analytical parameters [10]. Safaei et al. [11] set the bitumen failure definition at the peak of the product between C and the number of loading cycles (N) versus N. This research team comprehensively analyzed dynamic shear modulus, phase angle, S, and C values related to foamed asphalt binder, warm mix additive (WMA) modified bitumen, and neat asphalt to propose the mentioned failure definition (peak of C × N). As a result, the researchers also proved the usefulness of utilizing the LAS test and S-VECD model to evaluate the bitumen fatigue performance. Wang et al. [12] proposed as a failure definition the peak of the product of shear stress (τ) and N. In this study an exhaustive analysis of total, stored, and released pseudo-strain energy (PSE), such as so phase angle, shear stress, shear strain, and pseudo-strain values were conducted to reach the above conclusion (peak of τ × N). Those results were obtained by testing crumb rubber (CR) and styrene-butadiene-styrene (SBS) modified bitumens, terpolymer-modified asphalt binder, SBS-modified bitumen, and neat asphalt. Moreover, it was proven that the maximum stored PSE was equivalent to the peak of τ × N. Hence the maximum stored PSE could be used as a failure definition as well, which includes the energy in the process to identify the failure point. Cao and Wang [10] recommended a new failure definition to improve the accuracy of the previous proposal. The authors carried out this study by utilizing neat asphalt and SBS-modified bitumen. In this research work the scholars proposed the peak of C^2^ × N × (1-C) as a new failure definition. To reach this conclusion the researchers also utilized the parameters mentioned above.

Generally, the failure criterion defines the correlation between two variables: the first is related to material response, and the second is associated with loading input [10]. A failure criterion is a useful tool that can identify the failure point under different test conditions from those utilized in the model characterization testing. Wang et al. [12] proposed as a failure criterion ($G^{R}$) the average rate of released PSE based on the total released PSE ($W_{r,sum}^{R}$) in terms of the sum of all released PSE ($W_{r}^{R}$) up to the failure occurrence. In this study, a unique relationship between $G^{R}$ and $N_{f}$ was found and was independent of loading history, hence it was an important relationship for a specific material. The specific bitumens used in this study were mentioned above. Cao and Wang [10] identified that under certain conditions, the $G^{R}$-$N_{f}$ relationship was poor. As a result, a power law between the sum of stored PSE ($W_{sum}^{R}$) and a variable named Straining Effort (SE) was proposed as a new failure criterion. SE defines the necessary amount of mechanical effort to deform and damage the bitumen up to failure. The researchers found a strong correlation between $W_{sum}^{R}$ and SE, which illustrates the usefulness of this failure criterion. The specific bitumens used in this study were mentioned above. Want et al. [1] proposed a new type of failure criterion (*G^R^*) that is the average rate of released PSE, but it is based on the total released PSE (TRPSE) in terms of the area under the released PSE curve up to the failure occurrence. In this study, authors utilized neat asphalt, SBS-modified bitumen, and high viscosity additive-modified asphalt binder. The researchers demonstrated a strong correlation between the new failure criterion $G^{R}$ and $N_{f}$. This fact proved the suitability of the new proposal.

This research team would like to point out that bitumen fatigue performance is just one of the key factors influencing asphalt mixture fatigue performance. The asphalt binder-aggregate bonding represents another important factor to evaluate for ensuring a proper asphalt concrete fatigue performance [13], but it is outside the scope of this study.

In the case of the measure of self-healing capacity, a group of researchers has added a rest period into the asphalt binder fatigue-resistance-analysis protocol (LAS test) to evaluate the bitumen self-healing capability. This property can help to close cracks on road surfaces; hence this fact ensures strength and stiffness increment, and as a result pavement fatigue life is prolonged. Xie et al. [14] was the first study that included a rest period in the LAS test protocol (LASH), and the experimental data were processed by utilizing the S-VECD model to assess the self-healing potential of neat asphalt and SBS-modified bitumen. This research work proposed a percent healing (%H_s_) parameter calculated from the obtained DCC, a curve that illustrates C–S relationship, as mentioned above. Besides, a rest-damage superposition principle to elaborate the %H_s_ mastercurve was introduced, to eliminate the effect of damage level and rest period on %H_s_. In this study, neat asphalt binder exhibited better %H_s_ than SBS-modified bitumen. Wang et al. [15] assessed the healing potential of neat and SBS-modified bitumen subjected to short and long-term aging by conducting the LASH test and correlated the asphalt binder chemical composition with healing capability. Researchers found that the aging process and SBS (slightly) generally reduced the bitumen healing capability, and the existence of light fractions, small molecules, and longer molecules (fewer branches) contribute to asphalt binder healing ability. Wang et al. [16] evaluated the neat and SBS-modified bitumen healing behavior by utilizing the LASH and also analyzed their microstructures. This study concluded that higher concentrations of saturates and aromatics (light/low polarity) fractions, or greater existence of small molecules augmented the bitumen healing ability. SBS-modified bitumen showed similar healing capability to neat asphalt. Aurilio et al. [17] appraised the healing capacity of a WMA-modified bitumen by utilizing a simplified LASH. In this study, the authors stated that aged neat bitumen showed higher restoration capacity than that of aged WMA-modified asphalt binder.

However, only a few research works have studied the effect of self-healing polymer on bitumen performance. Two of those studies are Aurilio [18] and Aurilio and Baaj [19] which promoted the elastomeric properties of a self-healing elastomer (SHE) modified bitumen. The authors utilized a simplified LASH test to analyze the performance of the mentioned modified asphalt binder. Researchers concluded that it was difficult to assess the SHE role on bitumen restoration capability because the values of this parameter were almost the same. Besides, sometimes this type of elastomer enhanced bitumen fatigue performance in terms of the number of loading cycles to failure.

Numerous research works have spent time to improve the accuracy of failure definition and failure criterion to simulate the performance of neat and modified bitumens. Moreover, other studies have spent effort to assess the self-healing capability of neat and modified asphalt binders. All those research works have utilized LAS (or LASH) and the S-VECD model. However, just a few studies have evaluated the effect of SHEs on bitumen fatigue performance, hence it is still unclear the role of this type of elastomer on asphalt binder behavior. Futhermore, no information has been found about whether the current failure definition and failure criterion can accurately accommodate the SHE-modified bitumen performance. This research team considers that spending time and effort on these topics can pave the way to significantly increase bitumen fatigue performance by fully understanding its self-healing mechanism. As a result, the main objectives of this study are as follows:

To assess if the current failure definition, failure criterion, and fatigue performance analysis can simulate the SHE-modified bitumen behavior. If some inadequacies are found, convenient adjustments will be proposed.To evaluate if the current percent healing analysis framework can suitably accommodate the healing capability of SHE-modified asphalt binder. If some inadequacies are found, convenient adjustments will be proposed.

## 2. Materials and Methods

### 2.1. Materials

#### 2.1.1. Neat Asphalt Binder

This study utilized the 70# neat asphalt binder (NA) from Jiangsu province. The physical properties of this asphalt binder are shown in Table 1. The standard tests in this table are Chinese specifications based on AASHTO and ASTM specifications. Test results in Table 1 show that the selected NA matches the required values. As well, this NA grade has the ability to effectively withstand the cyclic load applied by the traffic [20]. Hence, research work is expected to be conducted successfully by selecting this NA.

#### 2.1.2. Self-Healing Elastomer

A room-temperature (25 °C) self-reinforcing self-healing thermoplastic polyurethane (STPU) was selected to be utilized in this research work to assess its effect on fatigue and healing properties of NA and evaluate its possible implementation as a bitumen modifier. At present, just a few studies have utilized this type of polymer for this target. This novel material was used to produce at the laboratory level the STPU modified bitumen (STPUB). STPU is in the developing stage, and it is the result of the collaboration between Zhengzhou and Sichuan Universities. The novelty of STPU is that a strain-induced crystallization was employed, to ensure a retarded but reversible self-reinforcing effect [21], this property could be useful and convenient for bitumens and asphalt mixtures. Moreover, this material does not need extra stimulus (microwave and induction heating) to promote self-restoration activity, which could be a valuable characteristic for the road surface. Table 2 shows the physical properties of STPU.

Materials used to synthesize the STPU: Polytetramethylene ether glycol (PTMEG, M_n_ = 1000 g/mol, *f* = 2), the catalyst dibutyltin dilaurate (DBTDL), and chain extender 3-Dimethylaminopropylamine (DMAPA) were purchased from Aladdin. Isophorone diisocyanate (IPDI) was purchased from Adamas. All of these chemical reagents were used without further purification. Tetrahydrofuran (THF, Sigma-Aldrich, St. Louis, MO, USA) and chloroform (CHCl_3_, Sigma-Aldrich) were used after CaH_2_ redistillation.

Synthesis of STPU: The synthetic pathway in this work includes two steps. The first step was to prepare prepolymer, during which the IPDI acts as chain extender because the R value (defined as the molar ratio of the isocyanate group to the hydroxyl group in PTMEG, [NCO]/[OH]) was deliberately set to 1.7. DMAPA as another chain extender was fed in the second step to consume residual NCO groups. In a typical procedure, PTMEG (10 g, 10.00 mmol) was first added into a four-necked flask equipped with a mechanical agitator and a mercury thermometer, which was heated by an electric heating jacket at 120 °C under a vacuum for 1 h to remove residual moisture and then cooled to 80 °C. IPDI (4.22 g, 19.00 mmol) was added into the flask and stirred for 1 h under an argon atmosphere, followed by adding DBTDL (1.26 mg, 2.0 × 10^−3^ mmol) dissolved in anhydrous CHCl_3_. The reaction continues at 80 °C for 3 h to obtain prepolymer. Thereafter, the temperature was cooled to 0 °C with iced brine. A total of 30 mL of anhydrous THF was charged into the system for tuning viscosity. DMAPA (0.92 g, 9.00 mmol) dissolved in THF (25 mL) was slowly added to the above solution by a constant pressure-dropping funnel within 30 min. The reaction mixture was continuously stirred at 0 °C for 1 h, then at room temperature for 2 h. Finally, the resulting viscous polymer solution was poured into rectangular molds and dried in an oven at 60 °C for 12 h, then desiccated in a vacuum oven at 80 °C for 24 h to remove residual solvent, resulting in elastomer film.

Figure 1 shows a summary of the synthesis process of STPU and its chemical structure. Figure 2 illustrates a piece of STPU as was received from Zhengzhou and Sichuan Universities and a piece of STPU split into small pieces (about 5 mm × 5 mm) to ease its mixture process with NA in (a) and (b) sections, respectively. For more information about the materials used to synthesize the STPU and its synthesis process, it is possible to consult Li et al. [21]. The main chemical characteristics of STPU are as follows: it is composed of a crystallizable soft segment (PTMEG) with a proper efficient length, which assures a lower crystallization energy threshold while conducting the elongation test. The presence of stratified H-bonding interactions is identified as sacrificial and dynamic bonds, which provide the hard domain with low binding energy. This fact ensures a crystalline configuration and an efficient vigorous exchange of H-bonds when the STPU is damaged [21].

### 2.2. Methods

#### 2.2.1. Preparation of STPUB

STPU is a material in the designing stage, hence there was no previous experience mixing this polymer with asphalt binder. As a result, it was necessary to conduct numerous trial and error tests to define the proper conditions to mix the NA and STPU. After this procedure, it was decided that 3500 rpm, 1 h, and 170 °C were the suitable shear speed, time, and temperature to obtain a homogeneous NA-STPU mixture, respectively.

The process to mix NA and STPU was as follows: The NA was heated in an oven up to 170 °C, in order to ensure a fluid state. Then, 1, 3, and 5 wt% of STPU were slowly added into the heated NA and the mixture was continuously stirred for 10 min by utilizing a glass rod. After this process, the mixtures were further sheared by utilizing a high-speed mixer (3500 rpm for 1 h), to ensure a homogeneous distribution of STPU in the NA. Modified bitumen containing 1, 3, and 5 wt% of STPU were defined as STPUB1.0, STPUB3.0, and STPUB5.0, respectively. In this study, the STPU contents in NA were set considering the polyurethane contents in previous research works, which ranged from 1% to 7% [22,23,24]. Hence, after numerous trial and error tests at the laboratory level, it was decided that STPU contents should be 1%, 3%, and 5%. Figure 3 shows the manually stirred process of STPU and NA, and the high-speed mixer blending both materials in (a) and (b) sections, respectively.

#### 2.2.2. Aging Procedure

The short-term aging of asphalt materials was conducted by using the Rolling Thin Film Oven (RTFO) test explained in the AASHTO T 240. The long-term aging of bitumen materials was carried out by utilizing the Pressurized Aging Vessel (PAV) test according to the AASHTO R28-12. NA, STPUB1.0, STPUB3.0, and STPUB5.0 were subjected to both RTFO and PAV processes, and then the specimens were tested according to LAS and LASH procedures.

#### 2.2.3. Performance Grade (PG) Characterization

PG characterization in this study was only conducted to define the fatigue and self-healing test temperatures. Flash point temperature test on unaged bitumens (AASHTO T48-06), rotational viscosity test on unaged asphalt binders (AASHTO T316), rutting index on unaged and RTFO aged bitumens (AASHTO T315-20), fatigue cracking index on RTFO + PAV aged bitumens (AASHTO T315-20), and Bending Beam Rheometer test on RTFO + PAV aged bitumens (AASHTO T313-12) were conducted to obtain the PG of NA, STPUB1.0, STPUB3.0, and STPUB5.0. The AASHTO M320-10 was employed to define the PG of each material used in this study. After finishing all procedures, the NA, STPUB1.0, STPUB3.0, and STPUB5.0 were classified as PG 64-22. Figure 4 illustrates DSR when conducting the rutting index test.

#### 2.2.4. LAS Test and S-VECD

The first step for conducting the LAS test is to know the linear viscoelastic behavior of asphalt materials. To this end, frequency sweep tests from 0.1 rad/s to 100 rad/s were carried out at different temperatures (15 °C, 25 °C, and 35 °C), and the strain level was set at 0.1%. The dynamic shear modulus results were utilized to obtain the master curve, to define the damage evolution rate “α”. This parameter was calculated as follows: α = 1/m + 1, where “m” is the higher slope (absolute value) in a log-log space of the obtained master curve [10,25]. Then, the LAS test (AASHTO TP101) was carried out, which is an effective test to assess the intermediate temperature fatigue performance of RTFO and RTFO + PAV aged specimens [26]. This procedure was conducted at 10 Hz with linear strain amplitude ramping from 0.1% to 30%, for 3100 cycles (standard) and it is defined as a continuous LAS test [14]. The cyclic strain rate (CSR) is the quotient between the highest strain and the number of cycles, for instance: standard CSR is 30%/3100 ≈ 0.01% per cycle. This parameter ranged from 0.005 to 0.02 because the loading cycles were set from 6200 to 1550, respectively, keeping a constant linear strain amplitude range (0.1% to 30%). The 8 mm parallel plate geometry and 2 mm gap were set in the DSR to evaluate the fatigue of RTFO + PAV-aged specimens. The temperature to undertake the standard LAS test was defined as follows: first, determine the average of low and high PG temperatures of the bitumen to be tested, second, add 4 °C to the obtained average [17]. Hence, the standard LAS test temperature was set at 25 °C. Table 3 shows the fatigue test matrix, and it was decided to include different CSRs and temperatures according to previous experiences [10,12].

Afterward, the S-VECD model was utilized to process the LAS test results. This model allows calculating the C and S, as mentioned above. The S-VECD model has a high efficiency because C and S relationship associated with bitumens are independently correlated to loading history, hence it can provide the possibility to determine any material fatigue response at any decided condition with limited experimental data [12,14,25]. The S-VECD model allows building the DCC, as mentioned before, which illustrates the correlation between C and S (see Equation (4)). In this research work, C and ΔS (damage increment) were calculated as follows [10]
(2)$$C=\frac{\left|G^{*}\right|}{{\left|G^{*}\right|}_{LVE}\cdot DMR}$$
(3)$$∆S_{i}={\left(\frac{1}{2}DMR\cdot {\left(\gamma_{i}^{R}\right)}^{2}\cdot \left(C_{i-1}-C_{i}\right)\right)}^{\frac{\alpha }{1+\alpha }}\cdot Q\;\mathrm{with}\;Q\equiv {\left[\int {\left(\sin\left(\omega_{r}\vartheta \right)\right)}^{2\alpha }d\vartheta \right]}^{\frac{1}{1+\alpha }}$$
(4)$$C=1-C_{1}*S^{C_{2}}$$where $G^{*}$, ${\left|G^{*}\right|}_{LVE}$, and DMR in Equation (2) are dynamic shear modulus (damaged), undamaged dynamic shear modulus measured in the linear viscoelastic range, and dynamic modulus ratio, respectively. Besides, $\gamma_{i}^{R}$, $\omega_{r}$, ϑ, and i-th in Equation (3) are pseudo-strain amplitude, reduced angular frequency, reduced time, and the cycle of interest, respectively. C_1_ and C_2_ in Equation (4) are regression constants. Parameters as stored pseudo-strain energy ($W^{R}$) and $\gamma^{R}$ were calculated as follows [10]:(5)$$W^{R}=\frac{1}{2}DMR\cdot C\left(S\right)\cdot {\left(\gamma^{R}\right)}^{2}$$
(6)$$\gamma^{R}\left(\vartheta \right)=\gamma \cdot {\left|G^{*}\right|}_{LVE}\cdot \sin\left(\omega_{r}\vartheta \right)$$where γ in Equation (6) is the shear strain amplitude.

Failure definition and failure criterion were also evaluated in this study. The former identifies the bitumen failure occurrence and hence defines its fatigue life, and the latter, most of the time, correlates two variables: the first related to material response and the second associated with the loading input [10], as explained before. The effectiveness of previously proposed failure definitions was evaluated (by utilizing LAS test results), for instance: the peak of C × N, the peak of τ × N [10,12], the maximum value of stored PSE [12] (see Figure 5) and the peak of C^2^ × N × (1-C) [10]. Furthermore, the usefulness of the formerly introduced failure criterion was assessed, and the equation is as follows [6]
(7)$$G^{R}=\beta {\left(N_{f}\right)}^{\partial }$$where β and *∂* are regression constants, $N_{f}$ and $G^{R}$ are the number of loading cycles to failure and the average rate of released PSE (failure criterion), respectively.

In this study, two variants of $G^{R}$ were calculated and evaluated: the first was determined by utilizing total released PSE ($W_{r,\;sum}^{R}$) in terms of the sum of all released PSE ($W_{r}^{R}$) (see Equation (8)) up to the failure occurrence, and the second was calculated by using total released PSE (TRPSE) in terms of the area under the released PSE curve up to failure occurrence (see Equation (10) and Figure 5).
(8)$$G^{R}\equiv \frac{W_{r,sum}^{R}}{{\left(N_{f}\right)}^{2}}\;with\;W_{r,\;sum}^{R}=\sum_{1}^{N_{f}}W_{r}^{R}$$
(9)$$W_{r}^{R}=\frac{1}{2}DMR*\;\left(1-C_{i}\right){\left(\gamma_{i}^{R}\right)}^{2}$$
(10)$$G^{R}=\frac{\bar{W_{r}^{R}}}{N_{f}}=\frac{TRPSE/N_{f}}{N_{f}}=\frac{TRPSE}{{\left(N_{f}\right)}^{2}}$$
(11)$$TRPSE=\int_{0}^{N_{f}}W_{r}^{R}dN=k\times {\left(\gamma \right)}^{2+2\alpha \left(\frac{C_{2}}{p}\right)}{\left(N_{f}\right)}^{\frac{C_{2}}{p}+1}$$
(12)$$k=\frac{1}{2}\times C_{1}\times {\left({\left|G^{*}\right|}_{LVE}\right)}^{2}\times q^{\left(-C_{2}/p\right)}\times \frac{1}{\left(C_{2}/p\right)+1}$$
(13)$$p=1-\alpha \times C_{2}+\alpha$$
(14)$$q=\frac{f\times 2^{\alpha }}{\left(1-\alpha \times C_{2}+\alpha \right){\left(C_{1}\times C_{2}\right)}^{\alpha }{\left({\left|G^{*}\right|}_{LVE}\right)}^{2\alpha }}\;\;\;f\to loading\;frequency$$

Another failure criterion recommended a power law relationship between total stored PSE ($W_{sum}^{R}$) and the Straining Effort (SE) [10].
(15)$$W_{sum}^{R}=K*SE^{\mu }$$where K and μ are regression constants, and SE represents the required mechanical effort to produce deformation and damage on the bitumen up to failure, and it was calculated as follows [10]:(16)$$SE\equiv \left(\sum_{i=1}^{N_{f}}\gamma_{i}\right)*{\left({\left|G^{*}\right|}_{LVE}\right)}^{2}$$
Figure 6 illustrates the 8 mm-bitumen specimens used in the LAS test. This type of asphalt binder samples was also used to conduct the LASH test.

#### 2.2.5. LASH Test and S-VECD

The LASH procedure is the LAS test with a rest period, as explained before. Different rest periods (RP) were set in the LASH test, for instance: 1 min, 5 min, 15 min, and 30 min. Moreover, 25% S_f_, 50% S_f_, 75% S_f_, 100% S_f_, and 150% S_f_ were the selected damage levels to apply the rest periods. The values for those parameters were decided considering previous experiences [14,17]. By setting these damage levels, it was possible to analyze the bitumen healing performance at the pre-failure, failure, and post-failure phases. The DSR setting conditions in LASH were the same as LAS, and the temperature was set at 25 °C.

Thixotropy is an important property of asphalt materials and should be taken into consideration to evaluate the self-healing performance of those materials [27]. At present, the LASH test measures restoration such as this term was defined by Leegwater et al. [28]. The restoration comprises both the “actual” self-healing as a reversible phenomenon and thixotropy. Considering this fact, in this study was used the term percent restoration (%R_s_) instead of percent healing (%H_s_) as proposed by Aurilio et al. [17]. %R_s_ was calculated as follows:(17)$$\%R_{s}=\frac{\left(S_{1}-S_{2}\right)}{S_{1}}$$where $S_{1}$ and $S_{2}$ are damage intensity at the end of the first loading phase and at the beginning of the 2nd loading phase, respectively. Figure 7 shows a schematic illustration of calculating %R_s_, continuous LAS test, and LASH test.

## 3. Results

### 3.1. Failure Definition Assessment

In this section, the peaks of C × N, τ × N (tau × N), C^2^ × N × (1-C), and the maximum values of stored PSE will be compared and analyzed. The latter parameter will be included in the analysis because it was proposed as a failure definition by Wang et al. [12] and this proposal was utilized to conduct the LASH test by Xie et al. [14]. Hence, it is convenient to include this parameter in the analysis. Besides, all failure definitions will be plotted on Shear stress vs. N, stored PSE vs. N, and DCC graphs, to assess its behavior in different types of graphs.

Figure 8 shows the failure definition points plotted on stress amplitude curves of asphalt materials. As can be seen from this figure, the stress curve related to STPUB5.0 is generally over the other curves on the crown regions, which demonstrates this modified bitumen shows a superior response (shear stress) than other asphalt materials in this study. Concerning the peak locations, those associated with STPUB5.0 exhibit higher $N_{f}$ (number of loading cycles to reach the failure point), regardless of the failure definition concept. These facts evidence the greater performance of STPUB5.0 than other bitumens, according to Figure 8. In the case of tau × N and maximum stored PSE failure definitions, their points are located close where the stress amplitude curves sharply drop, regardless of the asphalt materials. Moreover, C^2^ × N × (1-C) failure definition points are located at the left side (quite far) from the top of the stress amplitude curves, regardless of the bitumens. These mentioned failure points are in unsuitable positions, because the former group considers the $N_{f}$ too long and the latter too short, according to Figure 8.

The C × N peaks (Figure 8) seem to illustrate more reasonable locations on stress amplitude curves than other failure definition points, regardless of the bitumens, considered Figure 8. On the one hand, C × N and C^2^ × N × (1-C) failure points linked with STPUB1.0 and STPUB3.0 are placed at the right side of those related to NA, which evidences these two modified bitumens exhibit higher fatigue performance than NA, according to Figure 8 and these two failure definitions. However, NA presents a greater response (shear stress) than those two modified bitumens. On another hand, tau × N and maximum stored PSE failure points associated with STPUB1.0 and STPUB3.0 are located on the left side of those related to NA, which proves these two modified asphalt binders have lower fatigue life than NA, according to Figure 8 and these two failure definitions. In this case, NA still has a higher response (shear stress). Hence greater shear stress as a response in asphalt binder fails to ensure higher fatigue life for this type of material considering the information from failure definitions and Figure 8. Furthermore, tau × N and maximum stored PSE failure points are located in quite similar positions on stress amplitude curves, regardless of the bitumen; and in the case of NA and STPUB1.0, those failure points are perfectly matched.

Figure 9 shows the DCCs, and the failure definition points corresponding to all bitumens and failure concepts. This figure illustrates that DCC linked with STPUB1.0 is permanently above the other DCCs, which demonstrates a higher fatigue performance of this bitumen than others. In the case of the failure definition points, those corresponding with STPUB1.0 exhibit greater S than those related to other bitumens, regardless of the failure concept. It means that STPUB1.0 needs to be subjected to a higher S to reach the failure state, regardless of the failure definition. Moreover, STPUB1.0 must be subjected to higher S to reach one specific C, with respect to other bitumens in this study, hence STPUB1.0 needs upper S to reduce its C to close to 0. These findings conflict with the conclusion from Figure 8, which is STPUB5.0 has higher fatigue performance than other bitumens. As a result, higher $N_{f}$ will not ensure superior fatigue performance according to Figure 8 and Figure 9. Besides, Figure 9 depicts that all modified bitumens exhibit superior fatigue performance than NA, regardless of the failure definition, which demonstrates not only the positive effect of STPU on NA fatigue properties but also Figure 9 again disagrees with Figure 8, because this latter figure shows higher $N_{f}$ for NA than STPUB1.0 and STPUB3.0, in the case of tau × N and maximum stored PSE. It is interesting to mention that DCCs related to STPUB3.0 and STPUB5.0 almost match each other while collapsing, this fact can be interpreted as there should be an optimum STPU content, and higher values may not ensure superior fatigue performance.

The S-VECD model and C–S relationship have demonstrated a high efficiency to correlate bitumen fatigue and pavement cracking performances [10,12,29]. Moreover, the S-VECD theory was established considering the pseudo-strain domain in the elastic-viscoelastic correspondence principle [8]. As a result, pseudo-quantities such as pseudo-strain, pseudo-strain energy, and pseudo-stiffness must be utilized instead of corresponding physical variables to analyze bitumen fatigue performance [9,10]. Pseudo-strain domain eliminates the bitumen time-dependence performance, consequently, the true evaluation of asphalt material damage behavior can be conducted [12]. By utilizing the pseudo-strain instead of stress, it is possible to avoid the bitumen hysteretic behavior related to viscoelasticity [11]. Hence, this research team considers that Figure 9 illustrates the actual fatigue behavior related to each asphalt material, thereby other figures should be consistent with this figure.

Figure 10 illustrates the failure definition points identified on W^R^ curves. It can be seen from this figure that the W^R^ curve related to STPUB1.0 shows a superior performance (stored PSE) concerning other W^R^ curves. W^R^ represents the bitumen’s ability to store further energy in form of loading amplitude (energy input) while conducting the LAS test, hence higher W^R^ evidences greater capacity to store loading amplitude and vice versa [25]. Therefore, Figure 10 demonstrates the superiority of STPUB1.0 in terms of energy input, and also this material should be in better condition to withstand the loading amplitude in the LAS test.

Failure definition points linked with STPUB1.0 (Figure 10) show the upper ability to store loading amplitude, although those linked with STPUB5.0 exhibit higher $N_{f}$, regardless of the failure concepts. It is needed to point out that at the peak of the $W^{R}$ curve (STPUB5.0), the $W^{R}$ curve (STPUB1.0) still has a superior capacity for energy input, although this latter bitumen is in a post-failure state. This fact can be interpreted as even though bitumen has higher $N_{f}$ (in terms of loading cycles to reach the failure point) respect to other asphalt binders, this condition cannot assure greater asphalt fatigue performance than other asphalt materials. As a result, this research team concludes that stored PSE ($W^{R}$) is a highly influencing factor in bitumen fatigue behavior, to be consistent with Figure 9. Consequently, higher $N_{f}$ does not always guarantee greater fatigue performance, at least for this type of modified bitumens. Hence, $N_{f}$ fails to be a reliable parameter to identify the bitumen with superior fatigue performance.

With respect to failure definition point locations (Figure 10), those related to tau × N and maximum stored PSE are placed at the peak of the W^R^ curves (which is a logical position), regardless of the bitumen. In the case of C × N and C^2^ × N × (1-C) failure definition, their corresponding points are placed in non-logical places. Subsequently, this latter group of failure concepts cannot properly accommodate the fatigue performance of asphalt materials in this study. Considering these results and previous explanation related to S-VECD and pseudo-quantities, this research team consider that the maximum stored PSE failure definition is the most convenient failure concept, at least according to the experimental results obtained in this study.

After analyzing the S values related to the peaks of maximum stored PSE failure definition and modified bitumens is possible to confirm that STPUB1.0, STPUB3.0, and STPUB5.0 showed 67.1%, 22.1%, and 40.4% of fatigue performance increment respect to NA. This conclusion can be drawn considering the damage intensity that those materials could withstand up to the failure point. In the case of the tau × N failure definition those modified bitumens showed 67.1%, 24.0%, and 42.3% of fatigue increment, following the same criterion.

### 3.2. Failure Criterion Assessment

In this section, the $G^{R}$ (related to $W_{r,\;sum}^{R}$ and TRPSE) and $W_{sum}^{R}$ failure criteria will be analyzed to evaluate their usefulness to simulate neat and modified bitumen fatigue behaviors in this study. The tau × N and maximum stored PSE failure definitions showed logical failure point locations on the W^R^ curves, in the previous section, hence these two concepts will be utilized in the process to evaluate the utility of the above-mentioned failure criteria. Failure definition points will be plotted on $G^{R}$ vs. $N_{f}$ and $W_{sum}^{R}$ vs. SE graphs. “Test description 1, 2, 3, 4, and 5” in Table 3 will be utilized in this section.

Figure 11 shows the $G^{R}$ ($W_{r,\;sum}^{R}$)—$N_{f}$ relationship considering tau × N and maximum stored PSE failure definitions related to each bitumen in this study. Figure 11a,b illustrate a strong correlation between $G^{R}$ and $N_{f}$, because R^2^ values are equal to or higher than 0.98, which demonstrates that this failure criterion effectively predicts the fatigue life of NA and STPUB1.0, regardless of the testing conditions and failure definitions. Figure 11c still shows acceptable $G^{R}$ ($W_{r,\;sum}^{R}$)—$N_{f}$ relationships because R^2^ values are still higher than 0.9, hence this failure criterion is still adequate to simulate the fatigue behavior of STPUB3.0, at least at the selected testing conditions and failure definitions. In the case of Figure 11d, both correlations are poor because R^2^ values are close to 0, hence this failure criterion is not efficient to simulate STPUB5.0 fatigue performance. Besides, the slopes of the fitting equation graphs related to Figure 11a–c are similar, which means that this failure criterion identifies an alike tendency how changes the average rate of released PSE up to the failure point in NA, STPUB1.0, and STPUB3.0 (although the corresponding $N_{f}$ are different) and the slope is descendent as normal. However, in the case of Figure 11d, the fitting equation graph slope shows an ascendant tendency which is uncommon behavior for this parameter. Previous findings prove that G^R^ ($W_{r,\;sum}^{R}$)—$N_{f}$ correlations fail to be an effective failure criterion to simulate the fatigue life, for any type of bitumens and test conditions. This finding is in line with the finding from [10].

Figure 12 illustrates the correlation between $G^{R}$ (TRPSE) and $N_{f}$ (considering tau × N and maximum stored PSE failure definitions) which is a failure criterion proposed by Wang et al. [1] to refine the $G^{R}$ ($W_{r,\;sum}^{R}$)—$N_{f}$ failure criterion. Figure 12 demonstrates the existence of a strong correlation between $G^{R}$ (TRPSE) and $N_{f}$, regardless of the bitumens, experimental conditions, and failure definitions, because all R^2^ values are over 0.97. This fact evidences the usefulness of this failure criterion to predict bitumen fatigue life under different test conditions, at least with those selected in this study. The combination of $G^{R}$ (TRPSE)—$N_{f}$ failure criterion and maximum stored PSE failure definition has R^2^ values equal or slightly lower (neglectable difference) than the combination of $G^{R}$ (TRPSE)—*N_f_* failure criterion and tau × N failure definition. From these results is hard to confirm that one combination is better than another one, but it is possible to conclude that both have a strong capacity to predict and accommodate the fatigue behavior of bitumens in this study with the selected conditions, and this conclusion agrees with Wang et al. [1]

Figure 13 presents the $W_{sum}^{R}$—SE failure criterion (considering tau × N and maximum stored PSE failure definitions) which was proposed by Cao and Wang [10] to solve some deficiencies in $G^{R}$ ($W_{r,\;sum}^{R}$)—$N_{f}$ failure criterion (ex: higher fatigue life with higher aging). This figure depicts the weak correlation between $W_{sum}^{R}$ and SE because R^2^ values are lower than 0.6, and in some cases, these values are close to zero, regardless of the failure definitions, experimental conditions, and bitumens. Besides, the slopes of the fitting equation graphs associated with STPUB3.0 have a descendent tendency, regardless of the failure definitions, which is inconsistent with the results from Cao and Wang [10]. Previous findings prove that $W_{sum}^{R}$—SE failure criterion fails to be efficient in predicting bitumen fatigue life under different test conditions (selected test setting).

### 3.3. Percent Restoration and Fatigue Analysis

In this section, the bitumen self-restoration capability will be analyzed by setting the experimental conditions explained in section “2.2.5 LASH test and S-VECD”. The $S_{f}$ will be determined by following the proposal from Xie et al. [14] based on the maximum stored PSE failure definition. Figure 14 shows the %Rs related to each bitumen at different damage levels ($S_{f}$) and rest periods (RPs). “Appendix A” which contain Appendix A show the bitumen DCCs linked with 25%, 50%, 75%, 100%, and 150% of $S_{f}$, respectively. Additionally, each figure includes 1, 5, 15, and 30 min of RP. The criteria to choose these conditions to conduct the LASH test are as follows: include damage levels at pre-failure, failure, and post-failure phases of all bitumens and at different rest periods to evaluate restoration and fatigue performance at the same time. Those criteria are based on previous experiences, for instance: Xie et al. [14], Wang et al. [15], Wang et al. [16], and Aurilio et al. [17].

As can be noticed from Figure 14, %Rs generally increases while increasing RP and decreasing the percent of $S_{f}$, regardless of the asphalt binder and the failure phase. At the post-failure phase, the %Rs abruptly diminishes, indicating that bitumen has a higher %Rs when it is damaged by micro-cracks before the failure point. This fact proves that $S_{f}$ is an important threshold not only to evaluate fatigue performance but also to quantify %Rs. It is important to highlight that all these findings are consistent with previous experience, for instance [14,15]. Commonly, modified bitumens have higher %Rs than NA, which indicates that STPU can promote the self-restoration ability in asphalt binders. In numerous cases, STPUB3.0 exhibits higher or slightly greater %Rs than other bitumens in this study, but this behavior cannot ensure better fatigue performance for this bitumen when analyzing Appendix A. These figures demonstrate that STPUB1.0 exhibits superior fatigue performance than other bitumens, regardless of the RP and percent of $S_{f}$. Although this bitumen only presents three times (75% Sf—RP1, 150% Sf—RP1, and 150% Sf—RP5) greater %Rs than other asphalt binders in Figure 14. This finding confirms the existence of conflict with previous studies [14,15,16] because it was believed that the higher %Rs, the better the fatigue performance. Hence, it is needed to propose a new procedure to address this issue.

#### New Parameters to Evaluate Bitumen Restoration and Fatigue Performance

%Rs is a parameter that evaluates bitumen restoration capacity, but it is unable to assess how this restoration affects bitumen fatigue performance. Figure 14 and Appendix A demonstrate that higher %Rs does not ensure superior bitumen fatigue performance. Accordingly, in this study, new parameters are proposed to evaluate %Rs and fatigue performance at the same time, to ensure consistency in experimental results (bitumen with higher percent restoration will exhibit higher fatigue performance). The first proposal is fatigue-potential performance in the LAS test up to $S_{f}$ ($\varphi^{p}$), which is the total area below the DCC up to $S_{f}$ (the area below the continuous LAS curve, see Figure 15) when conducting the LAS test. The 2nd proposal is fatigue-potential performance in LASH up to $S_{f}^{\prime }$ (see Figure 15) ($\varphi^{ph}$), which is the total area below the DCC up to $S_{f}^{\prime }$ when conducting the LASH. In the case of LASH, the total area is the sum of the area below DCC before (area below the blue-circle defined curve, see Figure 15) and after (area below the red-circle defined curve, see Figure 15) the rest period. These two mentioned parameters will be used when evaluating the bitumen restoration up to 100% of $S_{f}$ (means that RP will be applied up to 100% of $S_{f}$.

Others two parameters will be proposed for bitumen restoration assessment at S values upper than $S_{f}$, which means RP will be applied at S values greater than $S_{f}$. The third proposal is fatigue-potential performance in the LAS test up to $S_{1}$ ($\varphi^{p\prime }$) (when $S_{1}$ > $S_{f}$, see Figure 16), which is the total area below the DCC up to $S_{1}$ (area below the blue-circle defined curve, see Figure 16) when conducting the LAS test. The 4th proposal is fatigue-potential performance in LASH up to $S_{1}^{\prime }$ ($\varphi^{ph\prime }$) (when $S_{1}$ > $S_{f}$, see Figure 16), which is the total area below the DCC up to $S_{1}^{\prime }$ when conducting the LASH. In this case, the total area is the sum of the area below DCC before (area below the blue-circle defined curve, see Figure 16) and after (area below the red-circle defined curve, see Figure 16) the rest period.

Percent restoration efficiency (%ξ) is another proposed parameter, which represents the percent increment of $\varphi^{ph}$ or $\varphi^{ph\prime }$ with respect to the $\varphi^{p}$ or $\varphi^{p\prime }$ of interest, respectively. The commonly used power law model to fit the DCC (Equation (4)) [10,25] was updated in this study, to obtain better agreement with the experimental results. The updated model (Equation (18)) was reached after numerous trial and error tests. Besides, it was proposed Equation (19), because it was identified that this equation had a superior agreement with the experimental results related to DCC after the rest period in the LASH test than Equations (4) and (18).
(18)$$C=1-C_{3}*\left(S^{C_{4}}+C_{5}\right)$$
(19)$$C=-log_{C_{6}}\left(S+C_{7}\right)+C_{8}$$where C_3_, C_4_, C_5_, C_6_, C_7_, and C_8_ are regression coefficients. Equations (20), (21), (22), (23) and (24) are defined (based on Equations (18) and (19)) to obtain $\varphi^{p}$, $\varphi^{ph}$, $\varphi^{p\prime }$, $\varphi^{ph\prime }$, and %ξ, respectively; where C_9_, C_10_, C_11_, C_12_, C_13_, C_14_, C_15_, C_16_, and C_17_ are regression coefficients.
(20)$$\varphi^{p}=\int_{0}^{S_{f}}1-C_{3}*\left(S^{C_{4}}+C_{5}\right)$$
(21)$$\varphi^{ph}=\int_{0}^{S_{1}}1-C_{9}*\left(S^{C_{10}}+C_{11}\right)+\int_{S_{2}}^{S_{f}^{\prime }}-\;log_{C_{6}}\left(S+C_{7}\right)+C_{8}$$
(22)$$\varphi^{p\prime }=\int_{0}^{S_{1}}1-C_{3}*\left(S^{C_{4}}+C_{5}\right)$$
(23)$$\varphi^{ph\prime }=\int_{0}^{S_{1}}1-C_{12}*\left(S^{C_{13}}+C_{14}\right)+\int_{S_{2}}^{S_{1}^{\prime }}-log_{C_{15}}\left(S+C_{16}\right)+C_{17}$$
(24)$$\%\xi =(\left(\varphi^{ph}-\varphi^{p}\right)/\varphi^{p})*100\;or\;\%\xi =(\left(\varphi^{ph\prime }-\varphi^{p\prime }\right)/\varphi^{p\prime })*100$$

See Figure 15 to understand Equations (20) and (21). See Figure 16 to understand Equations (22) and (23).

In this study, $\varphi^{p}$ and $\varphi^{p\prime }$ measure the bitumen fatigue performance when conducting the LAS test, in terms of the area below the DCC up to $S_{f}$ and after $S_{f}$ respectively, as explained above. This fact means the higher values of $\varphi^{p}$ and $\varphi^{p\prime }$, the higher position of DCC, hence superior bitumen fatigue performance. Moreover, $\varphi^{ph}$ and $\varphi^{ph\prime }$ measure the asphalt binder fatigue performance when carrying out the LASH test, in terms of the area below the DCC up to $S_{f}^{\prime }$ and up to $S_{1}^{\prime }$, respectively, as described before. Both areas include the bitumen restoration activity through the RP, and a higher area (related to $\varphi^{ph}$ and $\varphi^{ph\prime }$) means that the asphalt binder in question was able to increase its fatigue performance, considering the restoration capability. Because $S_{f}^{\prime }$ and $S_{1}^{\prime }$ are the damage intensity values associated with material integrity values $C_{f}$ and $C_{1}$ (after the RP), respectively. In other words how the bitumen can increase the damage intensity linked with the reference material integrity ($C_{f}$ and $C_{1}$) after the RP. Hence, greater values of $\varphi^{ph}$ and $\varphi^{ph\prime }$ means superior fatigue performance including the asphalt binder restoration activity. Accordingly, a higher %ξ value illustrates higher fatigue performance (including restoration capability) concerning the fatigue-potential performance of interest in the LAS test.

Figure 17 shows the %ξ of NA, STPUB1.0, STPUB3.0, and STPUB5.0 with respect to NA, which means the percent increment of $\varphi^{ph}$, and $\varphi^{ph\prime }$ linked with NA, STPUB1.0, STPUB3.0, and STPUB5.0 with respect to $\varphi^{p}$ and $\varphi^{p\prime }$ (related to NA), respectively. To evaluate fatigue performance and restoration capability at the same time of NA, STPUB1.0, STPUB3.0, and STPUB5.0 with respect to the NA, which is the original bitumen.

Figure 17 illustrates that STPUB1.0 has a higher %ξ with respect to NA than other bitumens in this study, which demonstrates this modified bitumen has superior fatigue performance including restoration capacity, regardless of the RP and damage level. This finding agrees with the conclusion from Appendix A, which are contained in the “Appendix A”. Generally, %ξ respect to NA increases while increasing the RP, regardless of the damage levels and bitumens; which proves RP is an influencing factor on bitumen fatigue performance. It is noteworthy to state that commonly, %ξ respect to NA increases from 0.25 to 0.75 S_f_ and decreases from 0.75 to 1.5 Sf, regardless RPs and bitumens. This fact means applying an RP (for restoration) at a lower damage level, does not ensure greater fatigue performance. Previously, in this research work and in previous studies [14,16] were found that the lower the damage level, the higher $\%R_{s}$ regardless of the RP and bitumen, which is in conflict with Figure 17. Additionally, higher $\%R_{s}$ in Figure 14 could not ensure higher fatigue performance in Appendix A. Hence, %Rs fails to be a useful parameter to assess bitumen fatigue behavior. The mentioned last finding from Figure 17 suggests that $S_{f}$ is a suitable threshold because 100% of $S_{f}$ normally stops the increment tendency of the %ξ respect to NA, regardless RPs and bitumens; indicating bitumen failure occurs. In Figure 17, all modified bitumens exhibit a superior %ξ than NA, which proves the effectiveness of STPU to improve the bitumen fatigue behavior; this finding agrees with Appendix A. The highest %ξ value in this study corresponds to STPUB1.0, which is equal to 154.02% increment. This result is associated with 0.75 S_f_ and RP equal to 30 min.

## 4. Discussion

The results related to C × N failure definition agree with previous research works [10,12] because in all cases the failure points are placed at the top of the stress amplitude curves, regardless of the bitumens. The results linked with τ × N and maximum stored PSE failure definitions also agree with the previous studies [10,12], because in previous and this studies the failure points are located close to where the stress amplitude curve sharply drops, regardless of the asphalt binder. Although the results linked with C × N, τ × N, and maximum stored PSE failure definitions are consistent with previous research works with respect to the failure point locations on stress amplitude curves, this fact by itself cannot proves these concepts are convenient to define the bitumen failure. Because stress amplitude curves failed to be consistent and reliable because, at higher shear stress linked with bitumen response, this material could not ensure higher fatigue performance. As a result, Want et al. [1] and Xie et al. [14] utilized the peak of the stored PSE curve as a failure definition, to define the bitumen fatigue life. This research team agrees with this consideration because stored PSE measures the material capacity to further store energy in form of loading amplitude (energy input) while conducting the LAS test [25], as explained before. The C^2^ × N × (1-C) failure point locations on stress amplitude curves are in conflict with the finding in the study of Cao and Wang [10], which evidences the inconsistency of this concept.

Moreover, this research team considers that $N_{f}$ only cannot decide which bitumen exhibits superior fatigue life consistent with the information from Figure 8, Figure 9 and Figure 10. Furthermore, stored PSE showed an evident influence on bitumen fatigue behavior according to the data from the above three figures. As a result, this research team considers that further experiments are necessary for studying the actual influence of stored PSE on asphalt binder fatigue performance. Hence, future research works will include those experiments.

The finding in this research work linked with $G^{R}$ ($W_{r,\;sum}^{R}$)—$N_{f}$ failure criterion conflicts with the results obtained in Wang et al. [12] and Safaei et al. [6] studies. These 2 research teams showed this failure criterion as a reliable concept to predict and simulate bitumen fatigue life at different test conditions, but the results in this study found a poor capacity of this concept in question to simulate the fatigue behavior of STPUB5.0, which demonstrates the inconsistency of $G^{R}$ ($W_{r,\;sum}^{R}$)—$N_{f}$ failure criterion. However, the finding (inconsistency) in this study agrees with the Cao and Wang [10] study, because these researchers identified that a high correlation between $G^{R}$ and $N_{f}$ fails to always ensure a proper relationship between $W_{r,\;sum}^{R}$ and $N_{f}$ which represents another inadequacy for $G^{R}$ ($W_{r,\;sum}^{R}$)—$N_{f}$ failure criterion. This fact proves this concept is not a reliable procedure to predict asphalt binder fatigue life under some specific conditions.

Moreover, the $G^{R}$ (TRPSE)—$N_{f}$ failure criterion results in this research work showed a high efficiency to predict and simulate the asphalt material fatigue life under different test conditions. This finding agrees with Want et al. [1] research work results. It is needed to point out that Want et al. [1] study utilized as failure definition the maximum stored PSE, but in this research work, the $G^{R}$ (TRPSE)—$N_{f}$ failure criterion was tested considering maximum stored PSE and tau × N failure definitions and both cases results were excellent. It is interesting to comment that under some conditions the tau × N concept slightly overcame the accuracy of the maximum stored PSE failure definition. As a result, this research team considers that it is needed to continue testing the $G^{R}$ (TRPSE)—$N_{f}$ failure criterion utilizing these two mentioned failure definitions under different experimental conditions to identify the combination to ensure better accuracy.

In the case of the $W_{\mathrm{sum}}^{R}$—SE failure criterion, the results obtained in this study disagree with the data shown by Cao and Wang [10]. These results were expected because the failure criterion in question was tested by utilizing the maximum stored PSE and tau × N failure definitions, but Cao and Wang [10] proposed this failure criterion based on the C^2^ × N × (1-C) failure definition. In Section 3.1 Failure definition assessment (in this study) was found a low accuracy of the C^2^ × N × (1-C) failure definition to identify bitumen fatigue life, hence it was decided to stop the analysis of this failure definition concept. As a result, this research team recommends avoiding the prediction of asphalt binder fatigue behavior by utilizing the $W_{sum}^{R}$—SE failure criterion and C^2^ × N × (1-C) failure definition.

In this study was found consistency and inconsistency with the current framework to evaluate bitumen fatigue and self-restoration capability. On the one hand, the consistencies were that %Rs generally increased while RP increased and the percent of $S_{f}$ decreased, regardless of the asphalt binder and the failure phase. Additionally, at the post-failure phase the %Rs abruptly diminished, indicating that bitumen failed, and $S_{f}$ was an important threshold not only to evaluate asphalt binder fatigue performance but also to quantify %Rs. Those findings agree with Xie et al. [14], Wang et al. [15], Wang et al. [16], and Aurilio et al. [17]. On another hand, the inadequacies were that bitumen with higher %Rs could not exhibit superior fatigue performance, and bitumen with a greater number of loading cycles to failure ($N_{f}$) was unable to show a better fatigue performance. These findings are in conflict with Xie et al. [14], Wang et al. [15], Wang et al. [16], Aurilio et al. [17], Want et al. [1], and Cao and Wang [10]. As a result, new parameters were proposed to assess the asphalt binder fatigue performance and self-restoration capability at the same time. The new procedure was consistent because the bitumen with higher percent restoration efficiency (%ξ) exhibited superior fatigue performance. Although these results, this research team considers that more experiments to validate the proposal of a new procedure to assess bitumen restoration and fatigue performance are needed. Hence, future research works will test the new procedure utilizing different test conditions and asphalt materials.

This team would like to point out some achievements which prove this study is different from previous research works that dealt with self-healing elastomers. The experimental results in this research study confirmed that a self-healing elastomer with suitable properties can always promote the self-restoration capability of bitumens and improve its fatigue performance. These conclusions have not been obtained in previous studies, for instance, Aurilio [18] and Aurilio and Baaj [19]. Moreover, the tests demonstrated that the current framework (LAS, LASH, and S-VECD) to evaluate asphalt binder self-restoration capability could not fully accommodate SHE-modified bitumens in this study. As a result, a new framework that considers self-restoration and fatigue performance at the same time was proposed.

It is noteworthy to mention that in a previous study sometimes que DCC after the rest period in LASH test collapsed with respect to continuous LAS curve [14], in this case the %ξ will be negative, which means the RP period was unable to increase or improve the fatigue performance of the bitumen in question.

## 5. Conclusions

This study analyzed the failure definition, failure criterion, fatigue performance, and %Rs testing neat and modified bitumens. The experimental procedure and the analysis were conducted by utilizing LAS, LASH, and S-VECD. In the case of LASH was considered different RPs and damage levels (pre-failure, failure, and post-failure phases) for having a wide idea about the fatigue behavior of bitumens in this research work. According to the experimental results, the following conclusions can be drawn:

Maximum stored PSE and τ × N failure definitions exhibited excellent capacities to accommodate the fatigue life of NA and SHE-modified bitumens. As well, both failure definitions identified STPUB1.0 with the highest fatigue performance increment (67.1%) with respect to NA.The $G^{R}$ (TRPSE)—$N_{f}$ failure criterion was the only one able to show a high efficiency (R^2^ up to 0.999) to predict and simulate asphalt binder fatigue life under different test conditions.Current restoration analysis framework was unable to fully accommodate %Rs and fatigue behavior of the SHE-modified asphalt binder.It was proved that higher %Rs did not ensure superior fatigue performance.It was proved that a greater number of loading cycles to failure ($N_{f}$) did not guarantee better fatigue performance.A new procedure (with new parameters) was proposed to assess the restoration and fatigue performance of asphalt materials, at the same time. It showed suitable results, but more experiments are needed. Besides, the new procedure identified STPUB1.0 with the greatest percent restoration efficiency up to 154.02%.According to the experimental data in this study, the SHE optimum content in NA was equal to 1%.Stored PSE is a factor with a high influence on bitumen fatigue performance.

## Figures and Tables

**Figure 1 materials-16-02488-f001:**
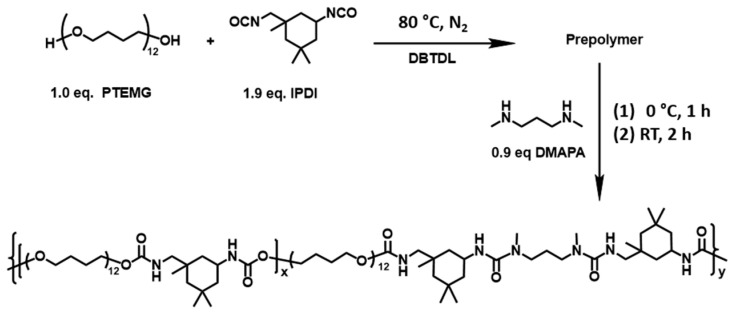
Synthesis of STPU and its chemical structure.

**Figure 2 materials-16-02488-f002:**
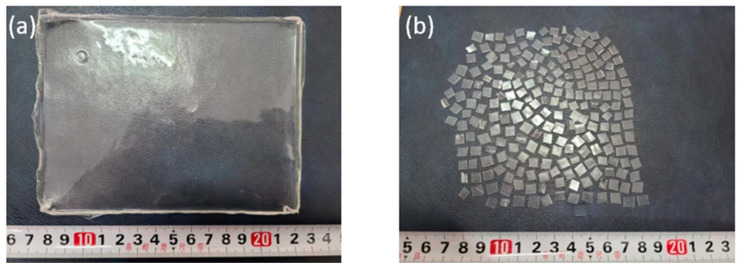
(**a**) A piece of STPU. (**b**) A piece of STPU splitted into small pieces (about 5 mm × 5 mm) [The rule in the picture is in centimeters].

**Figure 3 materials-16-02488-f003:**
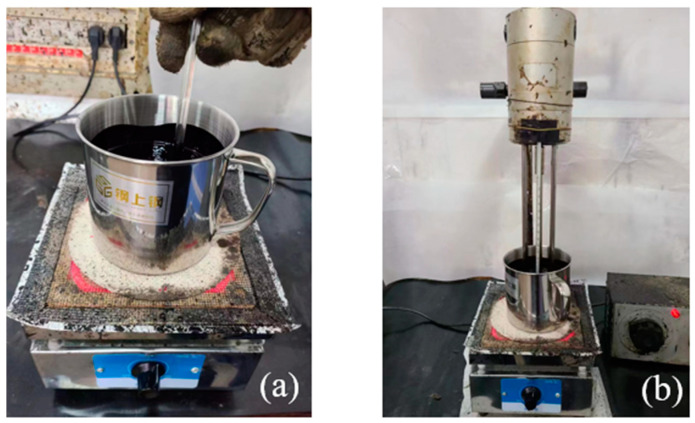
(**a**) Manually stirred process of STPU and NA. (**b**) High-speed mixer blending STPU and NA.

**Figure 4 materials-16-02488-f004:**
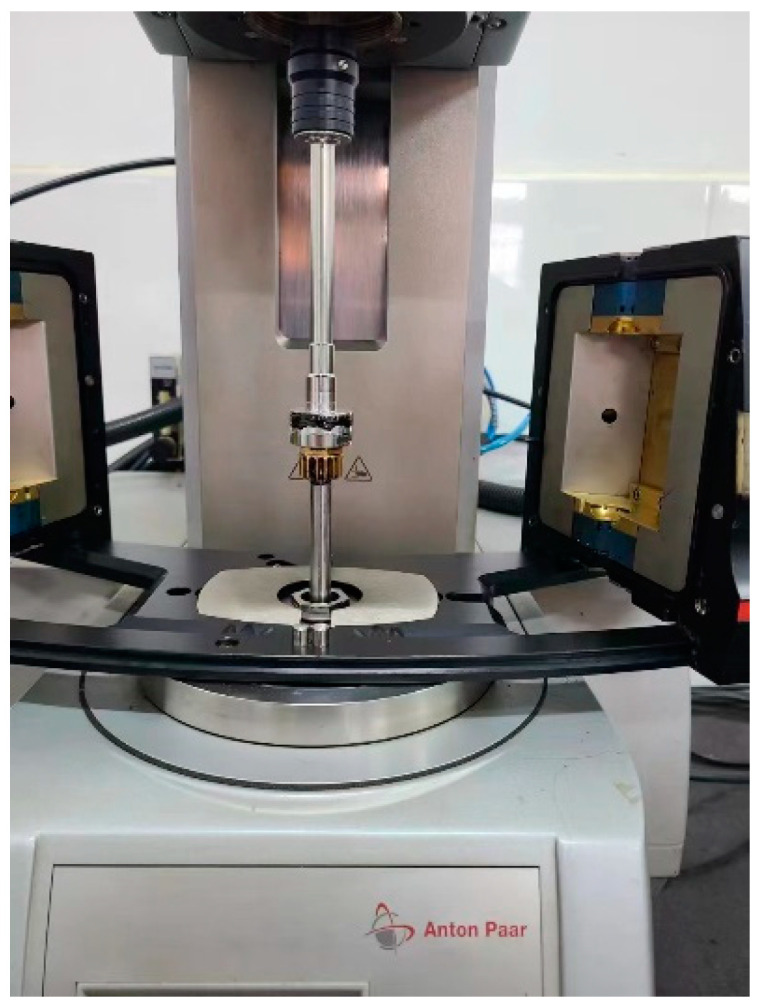
DSR when conducting the rutting index test.

**Figure 5 materials-16-02488-f005:**
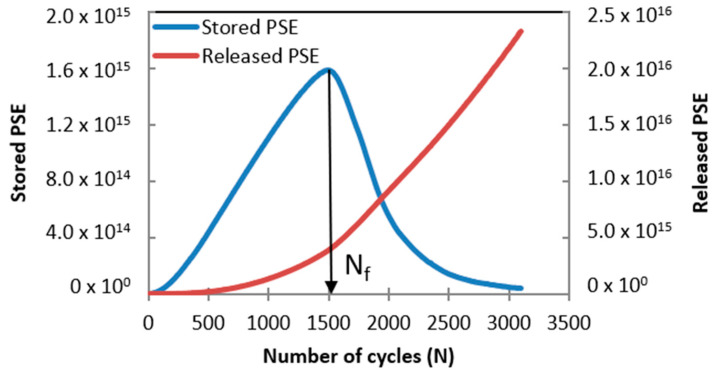
PSE-based failure definition.

**Figure 6 materials-16-02488-f006:**
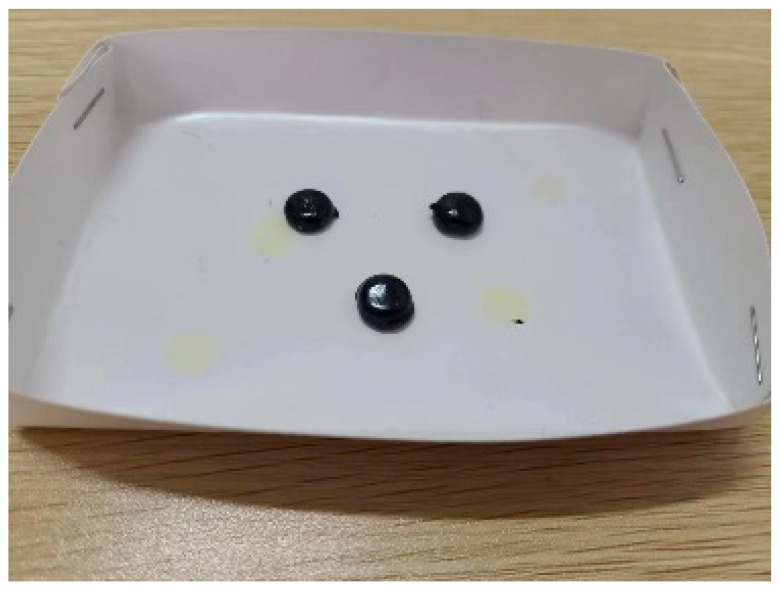
The 8 mm-bitumen specimens used in the LAS and LASH tests.

**Figure 7 materials-16-02488-f007:**
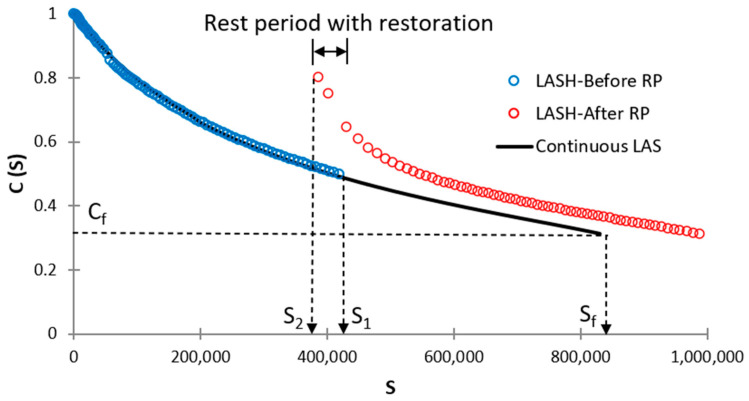
Schematic illustration of continuous LAS test, LASH test and calculating %Rs.

**Figure 8 materials-16-02488-f008:**
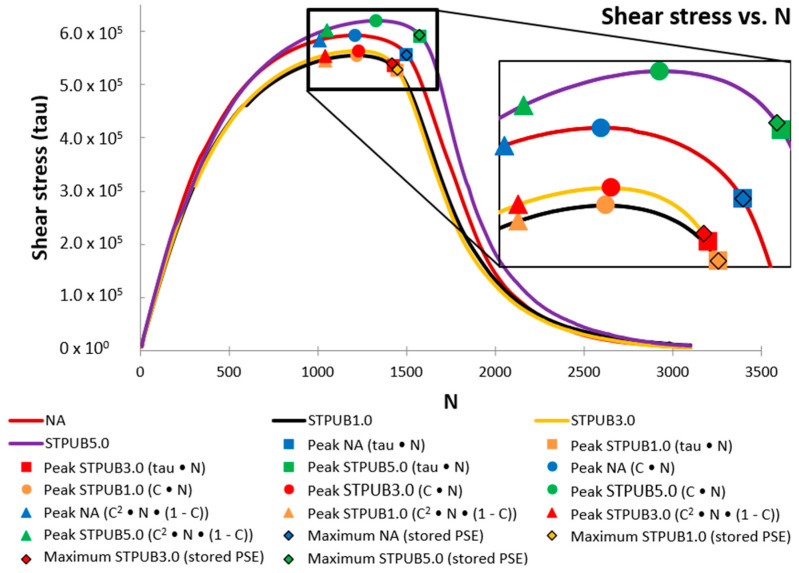
Failure definition points defined on the stress amplitude curve.

**Figure 9 materials-16-02488-f009:**
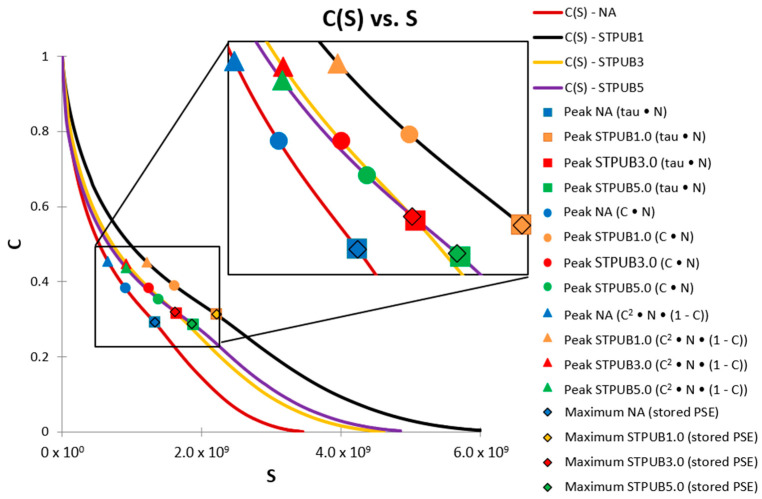
Failure definition points plotted on DCCs.

**Figure 10 materials-16-02488-f010:**
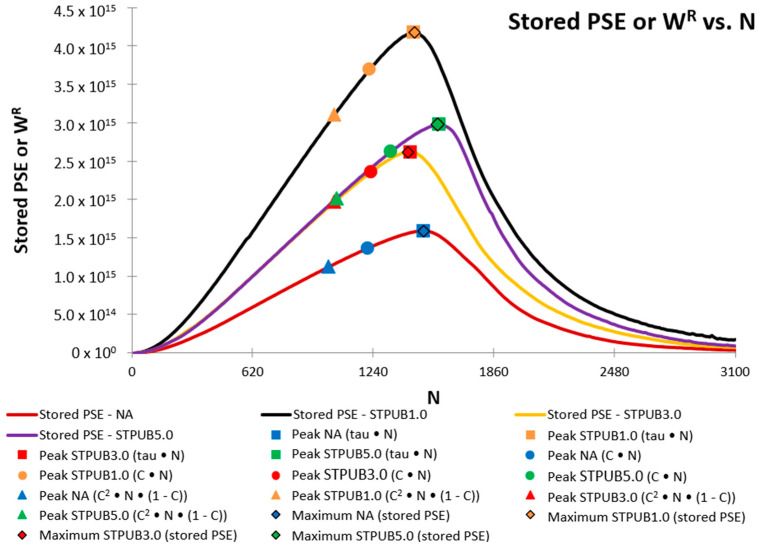
Failure definition points identified on stored PSE (WR) curves.

**Figure 11 materials-16-02488-f011:**
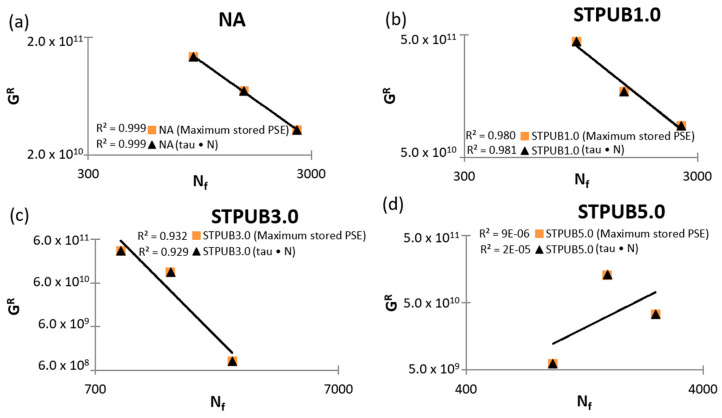
G^R^ ($W_{r,\;sum}^{R}$) vs. $N_{f}$ failure criterion graphs: (**a**) NA, (**b**) STPUB1.0, (**c**) STPUB3.0, and (**d**) STPUB5.0.

**Figure 12 materials-16-02488-f012:**
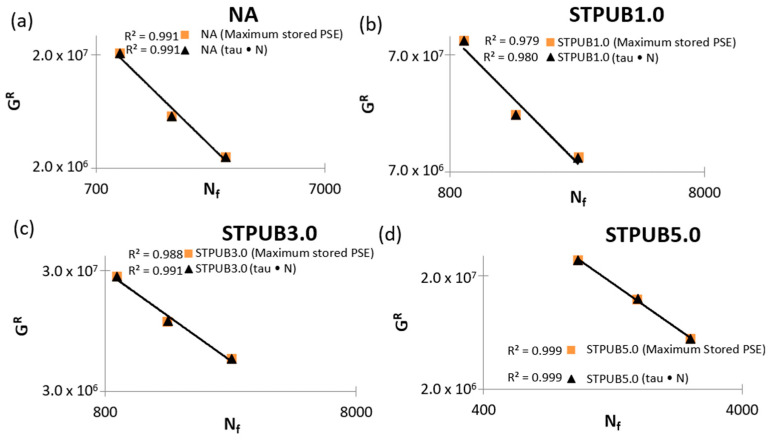
$G^{R}$ (TRPSE) vs. $N_{f}$ failure criterion graphs: (**a**) NA, (**b**) STPUB1.0, (**c**) STPUB3.0, and (**d**) STPUB5.0.

**Figure 13 materials-16-02488-f013:**
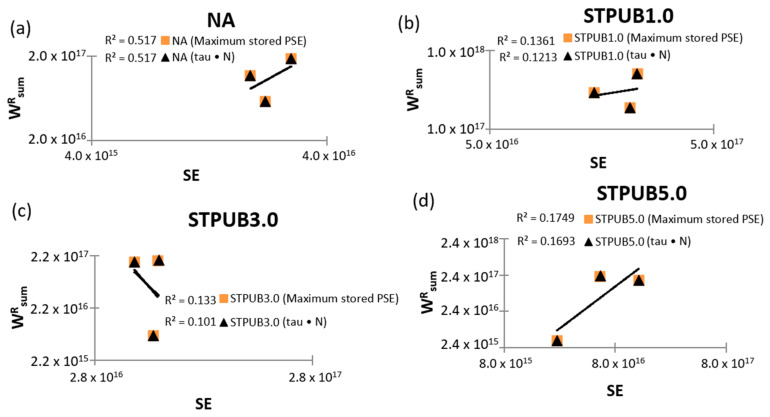
$W_{sum}^{R}$–SE failure criterion graphs: (**a**) NA, (**b**) STPUB1.0, (**c**) STPUB3.0, and (**d**) STPUB5.0.

**Figure 14 materials-16-02488-f014:**
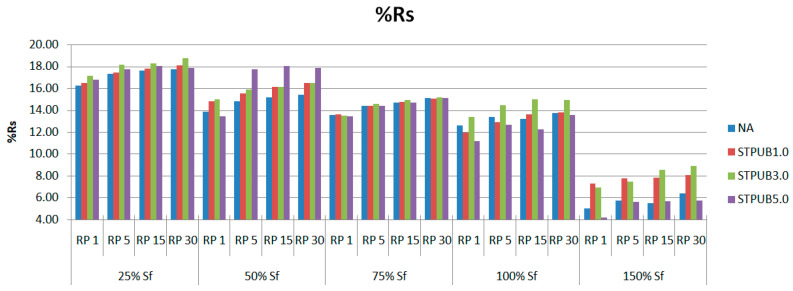
Percent restoration (%Rs) related to each bitumen at different damage levels and RPs.

**Figure 15 materials-16-02488-f015:**
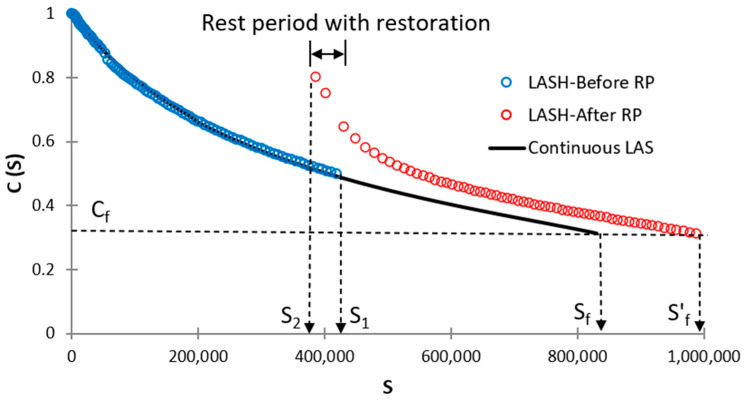
Schematic representation to identify $S_{f}^{\prime }$ and corresponding areas when conducting LASH (RP before or at $S_{f}$).

**Figure 16 materials-16-02488-f016:**
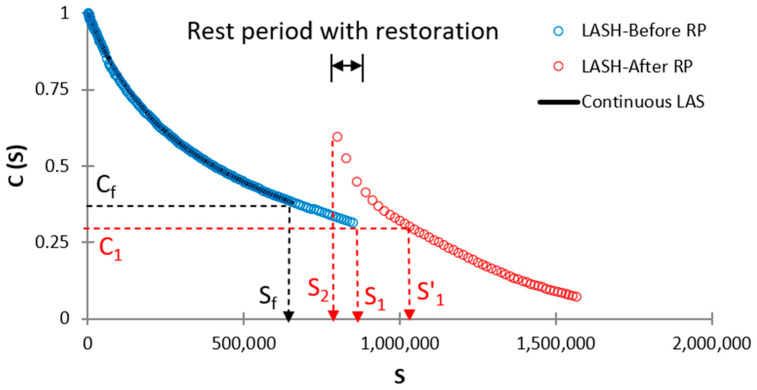
Schematic representation to identify $S_{1}^{\prime }$ and corresponding areas when conducting LASH (RP after $S_{f}$).

**Figure 17 materials-16-02488-f017:**
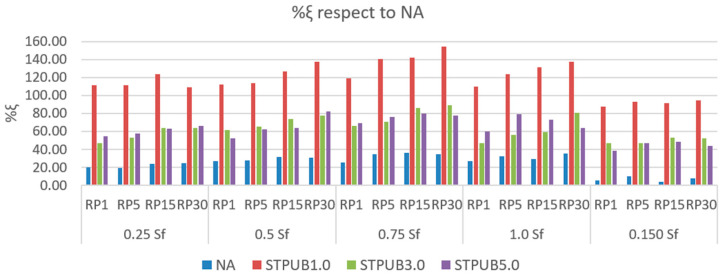
%ξ respect to NA with different RPs and damage levels.

**Table 1 materials-16-02488-t001:** Physical properties of 70# neat asphalt binder.

Test	Standard Value	Measured Value	Standard Test *
Penetration (25 °C, 5 s, 100 g) (0.1 mm)	60~80	60.1	T0604
PI	−1.5~1.0	−0.49	T0604
Softening point (°C)	≥46	51.1	T0606
Viscosity (60 °C) (Pa·s)	≥180	219	T0620
Ductility (10 °C)	≥45	62	T0605
Ductility (15 °C)	≥100	115.1	T0605
Wax content (%)	≤2.2	1.8	T0615
Flash point (°C)	≥260	300	T0611
Density (15 °C) (g/cm^3^)	-	1.033	T0603
After RTFO:			
Mass change (%)	≤±0.8	0.021	T0609
Residual penetration ratio (%)	≥61	67	T0604
Residual ductility (10 °C)	≥0.6	8	T0605

* Test methods are from Standard Test Methods of Bitumen and Bituminous Mixtures for Highway Engineering (JTG E-20-2011).

**Table 2 materials-16-02488-t002:** Physical properties of STPU.

Parameters	STPU Values
Tensile strength (MPa)	13.5 ± 2.2
Elongation [dried state] (%)	1460 ± 87
Density (g/cm^3^)	1.07
Melting point (°C)	120 ^a^

^a^ = obtained from temperature sweeping of rheological test.

**Table 3 materials-16-02488-t003:** Fatigue test matrix.

Material Information	Aging Condition	Test Description	Note
NA, STPUB1.0, STPUB3.0, STPUB5.0	PAV	1- LAS *: CSR ≈ 0.01%/cycle, 25 °C	To assess the effectiveness of failure definition
NA	PAV	2- LAS: CSR ≈ 0.01%/cycle, 25 °C CSR ≈ 0.005%/cycle, 25 °C CSR ≈ 0.02%/cycle, 23 °C	To assess the effectiveness of failure definition and failure criterion.
STPUB1.0	PAV	3- LAS: CSR ≈ 0.01%/cycle, 25 °C CSR ≈ 0.005%/cycle, 25 °C CSR ≈ 0.02%/cycle, 23 °C	To assess the effectiveness of failure definition and failure criterion.
STPUB3.0	PAV	4- LAS: CSR ≈ 0.01%/cycle, 25 °C CSR ≈ 0.005%/cycle, 27 °C CSR ≈ 0.02%/cycle, 25 °C	To assess the effectiveness of failure definition and failure criterion.
STPUB5.0	PAV	5- LAS: CSR ≈ 0.01%/cycle, 25 °C CSR ≈ 0.005%/cycle, 23 °C CSR ≈ 0.02%/cycle, 27 °C	To assess the effectiveness of failure definition and failure criterion.

* Standard LAS test.

## Data Availability

Not applicable.

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
