# Peer review of "Performance Assessment of Self-Healing Polymer-Modified Bitumens by Evaluating the Suitability of Current Failure Definition, Failure Criterion, and Fatigue-Restoration Criteria"

_materials, 2023, doi:10.3390/ma16062488_

Round 1

Reviewer 1 Report

The paper has an interest topic and it is well written enought, but sometimes the research objective is lost. The various topics covered lead to some confusion in the reading. In future, it might be useful for smoother reading to select the most important passages and findings.

Reviewer 2 Report

Following are my comments

1. Add some quantitative results in the abstract

2. There is lack of research gap/innovation in this research. Already studies have been conducted on using self-healing materials in binder/mix modification. Clearly explain, how this research is different than other studies.

3. Provide pictures of sample preparation and experimental tests

4. It is important to outline the importance of fatigue evaluation in asphalt mixture using ITFT or Beam fatigue test. Although it is out of the scope of this research, however, it should be summarized in the "Introduction". Because it is not only the bitumen but also the bitumen-Aggregate bonding performance that could lead to fatigue failure. e.g., these articles can be included in the literature "Fatigue Prediction Model and Stiffness Modulus for Semi-Flexible Pavement Surfacing Using Irradiated Waste Polyethylene Terephthalate-Based Cement Grouts", and Effective use of recycled waste PET in cementitious grouts for developing sustainable semi-flexible pavement surfacing using artificial neural network (ANN)", 

5. Why mixing of binder was performed at 3500 rpm for 1 hour? Provide justification.

6. provide the most significant quantity results in the conclusion.

7. Write the optimum value of SHE, that is achieved in this research

Reviewer 3 Report

The paper deals with a complex analysis on the use of an innovative polymer-modified bitumen and its effects on fatigue properties. The investigated topic is really interesting and the investigation was quite extensive and complete. Detailed explanations of the adopted procedures and obtained results are clearly highlighted.

The only (small) limitations can be found in the literature review and related references: although quite extensive and recent, some very important articles for the topic presented seem to be missing, especially in the field of self-repair properties. Some examples: “Testing Methods to Assess Healing Potential of Bituminous Binders (DOI: 10.1007/978-3-030-46455-4_7)”, “Evaluation of healing potential of bituminous binders using a viscoelastic continuum damage approach (DOI: 10.1016/j.conbuildmat.2018.05.228).

The supplementary materials is interesting and useful for the comprehension of the article.
